# Modification of Zirconia Implant Surfaces by Nd:YAG Laser Grooves: Does It Change Cell Behavior?

**DOI:** 10.3390/biomimetics7020049

**Published:** 2022-04-20

**Authors:** Mariana Brito da Cruz, Joana Faria Marques, Ana Filipa Silva Marques, Sara Madeira, Óscar Carvalho, Filipe Silva, João Caramês, António Duarte Sola Pereira da Mata

**Affiliations:** 1Oral Biology and Biochemistry Research Group, Unidade de Investigação em Ciências Orais e Biomédicas (UICOB), Faculdade de Medicina Dentária, Universidade de Lisboa, 1000 Lisboa, Portugal; jmarques2@campus.ul.pt (J.F.M.); afsm@campus.ul.pt (A.F.S.M.); admata2@campus.ul.pt (A.D.S.P.d.M.); 2Center for Microelectromechanical Systems (CMEMS), Department of Mechanical Engineering, University of Minho, 4800 Guimarães, Portugal; saramadeira@dem.uminho.pt (S.M.); oscar.carvalho@dem.uminho.pt (Ó.C.); fsamuel@dem.uminho.pt (F.S.); 3Bone Physiology Research Group, LIBPhys, Laboratory for Instrumentation, Biomedical Engineering and Radiation Physics, Faculty of Dental Medicine, Universidade de Lisboa, 1000 Lisboa, Portugal; carames@campus.ul.pt; 4CEMDBE–Cochrane Portugal, Universidade de Lisboa, 1000 Lisboa, Portugal

**Keywords:** laser, osteoblasts, fibroblasts, zirconia

## Abstract

The aim of this study was to evaluate gingival fibroblasts and human osteoblasts’ response to textured Nd:YAG laser microgrooves, with different dimensions, on zirconia implant surfaces. A total of 60 zirconia disks (8 mm in diameter and 2 mm in thickness) were produced and divided between four study groups (N = 15): three laser-textured (widths between 125.07 ± 5.29 μm and 45.36 ± 2.37 μm and depth values from 50.54 ± 2.48 μm to 23.01 ± 3.79 μm) and a control group without laser treatment. Human osteoblasts and gingival fibroblasts were cultured on these surfaces for 14 days. FEG-SEM (Field Emission Gun–Scanning Electron Microscope) images showed cellular adhesion at 24 h, with comparable morphology in all samples for both cell types. A similar cell spreading within the grooves and in the space between them was observed. Cell viability increased over time in all study groups; however, no differences were found between them. Additionally, proliferation, ALP (Alkaline phosphatase) activity, collagen type I, osteopontin and interleukin levels were not significantly different between any of the study groups for any of the cell types. Analysis of variance to compare parameters effect did not reveal statistically significant differences when comparing all groups in the different tests performed. The results obtained revealed similar cell behavior based on cell viability and differentiation on different microtopographic laser grooves, compared to a microtopography only established by sandblasting and acid-etching protocol, the reference surface treatment on zirconia dental implants.

## 1. Introduction

An ideal implant surface should present excellent osteoconductivity and biocompatibility. These features enhance the healing and osteogenesis of the peri-implant wound, leading to faster and more predictable osseointegration [1,2,3]. Additionally, soft tissue integration has been regarded as being of paramount importance in implant success since a good soft tissue seal has been referred to as essential to prevent and reduce the impact of peri-implantitis, which constitute one of the main factors in implant loss [4,5,6]. Peri-implant healing and accelerated osseointegration would allow clinicians to restore dental implants earlier, as well as provide more predictable results.

Yttria-stabilized polycrystalline tetragonal zirconia (Y-TZP) has been widely used in dental implants due to its esthetic and mechanical results [7,8] and has also been associated with a significant reduction in the formation of oral biofilms [9,10,11,12,13]. Zirconia implants, as stated in recent systematic and critical reviews, are described as a promising alternative to Titanium implants, for showing similar results in the clinical parameter of direct contact surface between bone and implant, as well as for adequate chewing load dissipation [14,15,16,17].

Several factors can influence the success of dental implants. Nevertheless, variations in surface parameters of zirconia dental implants, as described in the literature, interfere in the interface between material and peri-implant tissues, and in cellular behavior, and can significantly influence the speed and resistance of osseointegration [18,19,20,21,22]. In addition, parameters such as surface topography and roughness are important factors in the process of cellular contact, growth, and osteogenic differentiation [23].

Several strategies have been used to try to produce optimal surface characteristics in zirconia, namely, the texturization of implant surfaces by an innovative laser technique due to its associated advantages [24,25,26]. This method allows for a controlled and programmed surface characterization [24]. It is also characterized by the possibility to modify the topography without changing the surface chemistry, and it allows for a fast and precise patterning with grooves, without needing direct contact with the surface, which reduces the contamination risk through a more defined topography and chemical composition, compared to the traditional method of sandblasting and acid-etching [26,27].

Smooth surfaces are described in the literature as favoring the adhesion of human oral fibroblasts, as well as the growth of soft tissues, while the rough surfaces are considered to enhance the adhesion of osteoblasts and bone proliferation [28]. It is noteworthy that there are few references [29] associated with the Nd:YAG laser texturization of zirconia implant surfaces and that evaluate the influence of different dimensions of textured grooves performed by this technique, so it still not possible to extrapolate the real influence of these texturizations on cellular behavior.

Therefore, the aim of this study was to evaluate human osteoblasts and gingival fibroblasts’ response to zirconia implant surfaces textured with grooves with different dimensions by Nd:YAG laser machining. The hypothesis to be tested was that laser texturing would improve cell behavior when compared to the most common sandblasting and acid-etched zirconia implant surfaces alone.

## 2. Materials and Methods

### 2.1. Substrates

A total of 60 zirconia samples were produced, with a diameter of 8mm and thickness of 2 mm, from a commercial yttria stabilized zirconia powder (3Y-TZP) with uniform dispersion of 3 mol% of yttrium (3YSB-E, Tosoh Corporation, Yamaguchi, Japan) and with a particle size of 40 nm (average cluster size of 60 μm). The production was based on a “cold pressing” technique. All samples, except control group (D), were groove-textured by laser Nd:YAG (by Sisma ©, Vicenza, Italy) to obtain predefined grooves. This laser had the following features: a peak power per pulse of 8.57 kW, an output power of 6 W, a focal spot size of aproximatly30 μm, and a pulse width of about 35 ns operated at the repetition rate of 20 kHz; these parameters were defined based on previous studies [6,30].

All samples were sintered in an oven (Zirkonofen 700, South Tyrol, Italy) and then cleaned with isopropyl alcohol using ultrasonic equipment. Subsequently, all discs were sandblasted for 30 s with particles of alumina (Al_2_O_3_) with an average size of 250 μm, at a pressure of 6 bar and a distance of 12 cm, and washed again with ultrasonic equipment and isopropyl alcohol, as described in previous studies [24,25,26]. After this step, each sample was immersed in hydrofluoric acid (48% HF) for 30 min at room temperature and cleaned again using ultrasonic equipment, where they were immersed in isopropyl alcohol for 5 min. Lastly, samples were ultrasonically cleaned with 100% ethanol and sterilized in the autoclave to carry out all biological tests. The final aspect and dimensions of samples from each group, after surface treatment, were observed and evaluated by SEM JSM-6010 LV (JEOL Ltd., Tokyo, Japan) with the purpose of control groove dimensions. Secondary electron images were acquired at 500x magnification, at an acceleration voltage of 10 kV. Atomic contrast images were obtained using a Backscattering Electron Detector (BSED), with an acceleration voltage of 15 kV. All protocols were calibrated in previous studies [6,30,31,32].

#### Surface Characterization

To measure groove dimensions obtained after surface treatment, FEG-SEM images were undertaken before cell cultures (Figure 1). Pre-defined dimensions were not attained in their entirety due to questions related to the procedure, so Table 1 present the results obtained for width and depth grooves from all study groups.

### 2.2. Cell Culture

Human Fetal Osteoblasts (hFOB 1.19) were purchased from ATCC^®^ (CRL-11372TM; American Culture Collection, Manassas, VA, USA). Cells were cultured at 37 °C, in an atmosphere of 5% CO_2_ and 98% humidity. Culture medium was composed of a mixture (1:1 *v*/*v*) of Dulbecco’s modified Eagle’s Medium (DMEM) (BiowhittakerTM, LonzaTM, Basel, Switzerland) and Ham’s F-12 Medium (Sigma-Aldrich^®^ 51651C, St. Louis, MO, USA) supplemented with 0.3 mg/mL of G418 (by InvivoGen, Toulouse, France) and 10% of Bovine Fetal Serum (by Biowest^®^, Nuaillé, France).

Human Gingival Fibroblasts (HGF hTERT; Applied Biological Materials Inc., Richmond, BC, Canada) were cultured under the same conditions of temperature, humidity, and atmospheric composition as hFOB, but with a distinct culture medium, composed of a mixture of DMEM (Dulbecco’s modified Eagle’s medium—Lonza^®^, Switzerland) supplemented with 10% Bovine Fetal Serum (Biowest^®^, Nuaillé, France) and 1% Penicillin with streptomycin (G255 Applied Biological Materials Inc., Richmond, BC, Canada).

When cells reached approximately 100% confluence, trypsinization–trypsin EDTA (Lonza, Veners, Belgium) was carried out according to the manufacturer protocol. After that, cells were seeded with a density of 1 × 10^4^ cells/mL in 48-well plates containing sterile discs (Corning^®^, Corning, NY, USA) and cultured with culture medium previously heated at 37 °C. All experiments were conducted using a 4th passage [6,33].

#### 2.2.1. Cell Viability and Proliferation Assay of Osteoblasts and Fibroblasts

Cell viability and proliferation were evaluated using a resazurin-based viability method—Cell-Titer Blue^®^ reagent (Promega, Madison, WI, USA)—according to the manufacturer protocol. After 1, 3, 7 and 14 days of culture, the conversion rate was measured as fluorescence intensity in arbitrary fluorescence units (AU). Fluorescence intensity was detected at excitation/emission wavelengths of 560/590 nm using a Luminescence spectrometer (PerkinElmer LS 50B, Waltham, MA, USA) [6,33].

#### 2.2.2. Cell Morphology of Osteoblasts and Fibroblasts

To determine cell morphology, osteoblasts and fibroblasts were observed after one day of culture. After being washed with phosphate buffer saline (PBS), samples were fixed with 1.5% *v*/*v* glutaraldehyde solution for 10 min and dehydrated with increasing concentrations of ethanol (70%, 80%, 90%, and 100%). Samples were immersed in Hexamethyldisilazane (HMDS) (440191 Aldrich Chemistry, Milwaukee, WI, USA) and then covered with gold by the sputtering method (LEICA EM ACE600, Vienna, Austria), with an ultra-thin film (15 nm) of Gold-Palladium (80–20% weight), using a high-resolution sputter coater (208HR Cressington Company, Watford, United Kingdom) coupled to a MTM-20 Cressington High Resolution Thickness Controller. Scanning Electron Microscope (SEM) JSM-6010 LV (JEOL Ltd., Tokyo, Japan) were performed at different magnifications (500, 2000, and 5000×), at an acceleration voltage of 10 Kv. Atomic contrast images were obtained using a Backscattering Electron Detector (BSED), at an acceleration voltage of 15 kV. Image analysis was performed by two calibrated researchers, focusing on cell morphology, spreading, and the establishment of early contact with materials.

#### 2.2.3. Alkaline Phosphatase (ALP) Activity

Alkaline phosphatase activity was assessed at 7 and 14 days of osteoblast culture, using a fluorometric enzyme assay (ab83371 ALP assay Fluorometric, Abcam, Cambridge, UK) according to manufacturer instructions. A standard curve was performed at each measurement to calculate enzymatic activity. Standards and samples were measured using a fluorescence spectrometer (PerkinElmer LS 45, Waltham, MA, USA) with a fluorescent intensity at excitation/emission wavelengths of 360/440 nm. Values were converted to mU/µL of enzyme (ALP) based on the standard regression equation.

#### 2.2.4. Quantification of Interleukin 1β (IL-1β) by ELISA Method

Quantification of interleukin IL-1ß was performed in osteoblast cell cultures at 1 and 3 days, using Human IL-1b Chemiluminescent ELISA Kit (LumiAB^TM^ EMELCA Bioscience, Clinge, The Netherlands) according to the manufacturer protocol. Samples were measured by a luminescence technique with Victor Nivo Multimode Plate Reader (PerkinElmer^®^ Inc., Waltham, MA, EUA). Results were obtained in units of absorbance (nm) and converted into pg/mL, by the correlation with the linear regression equation obtained in the calibration curve.

#### 2.2.5. Quantification of Collagen by ELISA Method

The quantification of collagen in each osteoblast and fibroblast cell cultures was performed at 3 and 7 days. HumanPro-Collagen I alpha 1 DuoSet Elisa Kit (DY6220 05 R&D Systems, Inc., Minneapolis, MN, USA) was used according to manufacturer protocol. All samples fluorescence intensities were detected at excitation/emission wavelengths of 450nm using Victor Nivo Multimode Plate Reader (PerkinElmer^®^ Inc., Waltham, MA, USA). Results were acquired as absorbance units (nm) relative to the relative light intensity and converted according to the standard curve performed in pg/mL.

#### 2.2.6. Quantification of Osteopontin by ELISA Method

The osteopontin level presented in the supernatant was determined using Human Osteopontin Chemiluminescent ELISA kit (LumiAB^TM^ EMELCA Bioscience, Clinge, The Netherlands) by technique of luminescence with Victor Nivo Multimode Plate Reader (PerkinElmer^®^ Inc., Waltham, MA, USA) at 3 and 7 days of osteoblasts culture. The fluorescence intensity of samples was measured at excitation/emission wavelengths of 700 nm. Results were obtained in absorbance units (nm) per relative light intensity values and converted accordingly through the standard curve to pg/mL.

#### 2.2.7. Quantification of Interleukin 8 by ELISA Method

Interleukin 8 quantification was performed at pre-defined time of 1 and 3 days of fibroblasts culture, using Human IL-8 Chemiluminescent ELISA kit (LumiAB^TM^ EMELCA Bioscience, Clinge, The Netherlands), a luminescence technique, that was read using a multimode microplate reader—Victor Nivo Multimode Plate Reader (PerkinElmer^®^ Inc., Waltham, MA, USA). Results were obtained in absorbance units (nm) and converted to pg/mL, through a standard curve.

### 2.3. Statistical Analysis

Statistical analysis was performed using IBM^®^ SPSS^®^ 24.0 software for Mac (SPSS, Chicago, IL, USA). The data were tested for normality (Shapiro–Wilk test). A comparison between groups for cell viability, proliferation, ALP, interleukin 1β, collagen, osteopontin, and interleukin 8 was performed based on a Kruskal–Wallis test, followed by Dunn’s test or Mann–Whitney with Tukey’s post-hoc tests as appropriate, to identify groups with significant differences. The significance level was set at *p* < 0.05. All data are presented as a mean ± standard deviation (SD).

Estimations of sample size were completed for each of the tests performed based on a type I error rate (α) of 0.05 and a statistical power of 0.80 (ß). Based on the preliminary results of cell viability [33] and with the aid of a simulator for calculating the size of the sample: G*Power 3.1.9.0 for Mac, (Dusseldorf, Germany), an N = 12 was determined for each group as needed to detect an effect dimension of 1.4 with a power statistical of 0.80. Due to protocol convenience, the total dimension of the sample was N = 15.

## 3. Results

### 3.1. Cell Viability and Proliferation Assay of Osteoblasts and Fibroblasts

The results of cell viability and proliferation were obtained for 1, 3, 7, and 14 days (the last one was only measured in osteoblasts) for the two different cell cultures: osteoblasts and fibroblasts (Figure 2).

Osteoblast’s viability increased over time in all study groups. However, no differences were observed between them (*p* > 0.05). Additionally, osteoblast proliferation was not significantly different between groups (*p* > 0.05).

Fibroblast’s behavior showed a similar pattern: no statistically significant differences were observed between study groups over culture time either in cell viability or proliferation (*p* > 0.05).

### 3.2. Cell Morphology

Sample images obtained by SEM after 1 day of culture of osteoblasts (Figure 3a) and fibroblasts (Figure 3b) are presented with the respective magnifications.

SEM images reveal adherent osteoblasts for all groups tested after just 1 day of culture. Osteoblasts cells had similar morphologies among all groups under study, with a more rounded cytoplasmic membrane and with less spreading of their cell bodies. A similar cell spreading within the grooves and in the space between them was observed. Thus, grooves’ texture by laser does not seem to influence osteoblasts distribution. However, cells inside grooves seem to be more rounded, while in the areas between them, cell bodies show greater spreading and cellular extensions. Cell distribution appears more homogeneous and the anatomy more regular over the entire surface of the control group.

After 1 day of culture, adherent fibroblasts were observed in all groups. Additionally, fibroblasts cells have similar morphology, although with a more rounded phenotype and without finding a true scattering of cell bodies characteristic of fibroblasts. Group C, which is the deepest group, but with smaller width, appears to have more elongated cell bodies, as in the control group, showing greater spreading and cell adhesion. Regarding the distribution and quantity of adhered fibroblasts, there seems to be no differences between the groups, with the exception of the control group, which seems to reveal a greater cellular quantity.

### 3.3. Alkaline Phosphatase Activity (ALP)

ALP results on osteoblast suspension at 7 and 14 days of culture are shown in Figure 4. The control group revealed the highest results of the concentration of alkaline phosphatase. However, when compared with all groups, the differences were not statistically significant (*p* > 0.05).

### 3.4. Interleukin 1β

The analysis of interleukin 1ß production by the osteoblast culture, in all study groups, was performed at 1 and 3 days of culture, as shown in Figure 5. The results obtained show similar values of IL 1ß concentration at 1 and 3 days of culture, in all study groups, without statistically significant differences between them in the two mentioned measurement times (*p* > 0.05).

### 3.5. Collagen

Collagen content in the extracellular medium was another parameter to be evaluated in cell cultures (Figure 6). This measurement was determined in osteoblast and fibroblast cultures at 3 and 7 days and culture.

No statistically significant differences were observed in both cultures at pre-defined timepoints, when compared study groups (*p* > 0.05).

### 3.6. Osteopontin

Osteopontin concentration in osteoblast cell culture was evaluated at 3 and 7 days of culture (Figure 7). At 3 and 7 days of culture, the A group revealed the highest concentration of osteopontin. However, when compared to the other groups, no statistically significant differences were found between them (*p* > 0.05).

### 3.7. Interleukin 8 (IL-8)

Interleukin 8 production by fibroblasts was shown in Figure 8. From 1 to 3 days, interleukin 8 concentration found in extracellular medium decreased in all groups. However, after statistical analysis, no significant differences were found between groups after 1 and 3 days of culture (*p* > 0.05).

## 4. Discussion

The literature presents zirconia implants as a promising alternative to Titanium. In this context, several modifications of zirconia implants’ surfaces have been carried out with the aim of improving their biological behavior [8,15,16,21]. In parallel, more recent studies are beginning to explore other innovative techniques, such as laser irradiation, in order to improve the surface treatment of zirconia [34,35]. In this study, we aimed to evaluate the influence on peri-implant cell behavior of laser-textured groves with different dimensions (controlled in post-production) on zirconia surfaces and to compare them with a control that had no laser texturing performed. This differs from previous studies [29] that have only looked into osteoblast behavior on sandblasted and microgrooved zirconia disks and compared it to behavior on only sandblasted surfaces and not sandblasted and acid-etched surfaces, which are considered the gold-standard surfaces; therefore, this study provides a unique look into the capabilities of adding laser texturing compared to just sandblasting and acid-etching. This study also provides novelty by including soft-tissue cells behavior—gingival fibroblasts—to assess the peri-implant soft tissue responses to these surfaces, which are critical for a successful outcome [4,5].

The results of our study reveal that, although a preliminary study [30] was carried out on the effect of the laser parameters to ascertain the best parameters to obtain the desired dimensions, the projected conditions revealed dispersion in the different parameters: widths from 45 μm to 130 μm and depths from 20 μm to 50 μm. Through the cross-sectional SEM images, it is possible to verify that grooves have a triangular cross-sectional shape, which results from successive laser passages with an increase in depth. However, different dimensions of width and depth were obtained by increasing laser beam speed. This speed increase results in smaller width and depth. Laser beam speed above 512 mm/s does not reproduce the desired line. Laser beam speed above 2–8mm/s leads to the superficial fusion of zirconia, resulting in thermal stress and, consequently, cracks [36]. Therefore, it is necessary to choose a laser speed intermediate so that none of these things occur.

The ideal dimension for maximizing osteointegration in relation to the appearance of or distribution on the implant surfaces remains under intense research [21,29]. In our study, SEM images after one day of culture reveal that microtextured surfaces with grooves by laser machining do not seem to influence the adhesion and distribution of human osteoblasts and human gingival fibroblasts. Cell distribution was similar inside the grooves and in the space between them, without a signal of changes in cell distribution according to the sample pattern under study. These results are contrary to those observed by Zhu and collaborators in 2005, who revealed that grooves and microtextured surfaces can provide orientational and directional clues, through a phenomenon known as “contact orientation” for osteoblast morphogenesis in the preferred direction [23]. Delgado-Ruiz et al. [29] also reported similar results, observing that microgrooves with 30 µm width have a significant influence on osteoblast morphology and alignment. These variations in results might be due to the varying dimension of the grooves, which may imply that larger grooves (within the dimensions used in our study) may lead to a less favorable outcome. Further studies should aim to explore this question and understand what actual dimensions are more favorable to have a favorable impact on cell viability and differentiation.

Regarding the effect of textured groove width by laser machining on cell viability and proliferation, when compared with the control surface (without laser texturing), no differences were found in the cellular response of human osteoblasts and human gingival fibroblasts, thus rejecting the original hypothesis. This result is in accordance with an in vitro study carried out by Holthaus and collaborators in 2012, in which the depth of the microchannel had virtually no significant impact on osteoblasts [37]. However, a study by Nadeem and collaborators on human mesenchymal cells revealed contrary results when they observed that grooves greater than 50 μm showed a more favorable osteogenic response when compared to grooves of 10 μm, but the protocol was performed in an alumina ceramic material, which may present different results from the zirconia used in the present study [38]. Additionally, Delgado-Ruiz and collaborators’ results published in 2016 [29] showed a significant difference in osteoblast cell viability and proliferation in microgrooved laser-textured zirconia in comparison to sandblasted surfaces, but as referred to previously, their control group was different and laser-textured surfaces were only sandblasted and textured with a femtosecond laser, which differs from the protocol used in this research work.

As for cell activity, ALP activity showed no significant differences between study groups, only having a non-significant increase in value in the control group. These results are not in agreement with the ones observed by Delgado-Ruiz and collaborators [29], where ALP production was increased on laser-textured surfaces, compared to sandblasted control. It is referred to in previous studies that ALP production increase is associated with microroughness values over 3 µm [39], and sandblasted and acid etched surfaces produce micro and nanotopographies that enhance cell proliferation and differentiation [40]; therefore, there were probably not enough differences in roughness between laser and just sandblasted and acid-etched samples to elicit a different cell response. Nevertheless, this relation needs further investigation to be fully understood.

Interleukin, collagen, and osteopontin production showed no significative differences between laser texturing and non-laser textured samples. These indicators of differentiation and inflammation (IL-8 is a potent chemotactic agent for neutrophils and IL-1 beta is a strong pro-inflammatory factor, leading to osteoclast activation and inflammatory-driven bone resorption) were similar between the reference surfaces, and surfaces that were laser-textured show adequate results for this type of texturization, suggesting that it does not change cell response; therefore, all the advantages of laser texturing can be used without increasing inflammation or decreasing cell differentiation [24,25,26,27].

The results presented are representative of the pattern created with Nd:YAG laser, which has some limitations compared to other existing laser techniques, such as the CO2 or femtosecond laser [38,41]. Therefore, the results obtained in our study may also be influenced by the type of laser used, as defended by Minamizato and his collaborators [42]. His study processed flat zirconia surfaces with pulsed Nd:YAG lasers to drill 500 μm diameter holes, through a 2 mm thickness in the intraosseous portion of the zirconia implants, and reported the presence of a layer of molten and solidified zirconia around the holes [42].

This is an in vitro study that is important to the knowledge and understanding of cellular mechanisms in osseointegration and soft tissue integration processes. However, these are highly complex processes that are impossible to fully model in an in vitro study. Therefore, these results need to be complemented with more complex in vitro and in vivo studies, including bacteriological studies, given that any surface developed, considering the current knowledge of osteointegration phenomena, only becomes innovative if it models and influences bacterial behavior, especially regarding biofilm formation dynamics.

## 5. Conclusions

This study reveals that adding microgrooves patterns with different dimensions from 45 µm to 125 µm by an Nd:YAG laser on conventional sandblasting and acid-etched zirconia implant surfaces does not influence the viability, proliferation, and inflammatory mediation response of human osteoblast and human gingival fibroblast.

## Figures and Tables

**Figure 1 biomimetics-07-00049-f001:**
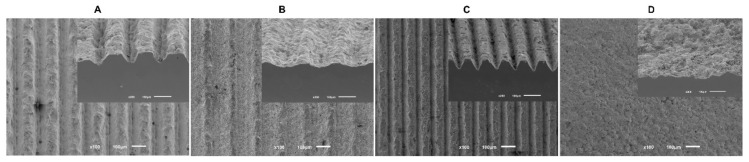
The FEG-SEM micrographs taken after the surface treatment of all samples (100× magnification).

**Figure 2 biomimetics-07-00049-f002:**
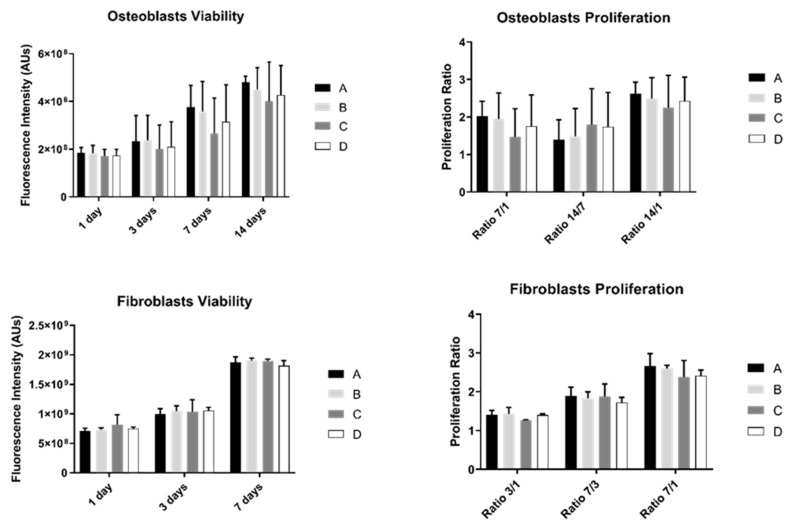
Graphs showing osteoblasts and fibroblasts’ viability as a mean of the registered fluorescence intensity of resorufin expressed in arbitrary units (N = 15) for each study group (as described in Table 1). Error bars represent a standard deviation. A Kruskal–Wallis, followed by Dunn’s test, was used for comparisons between study groups. Statistical significance was defined as: *p* < 0.05.

**Figure 3 biomimetics-07-00049-f003:**
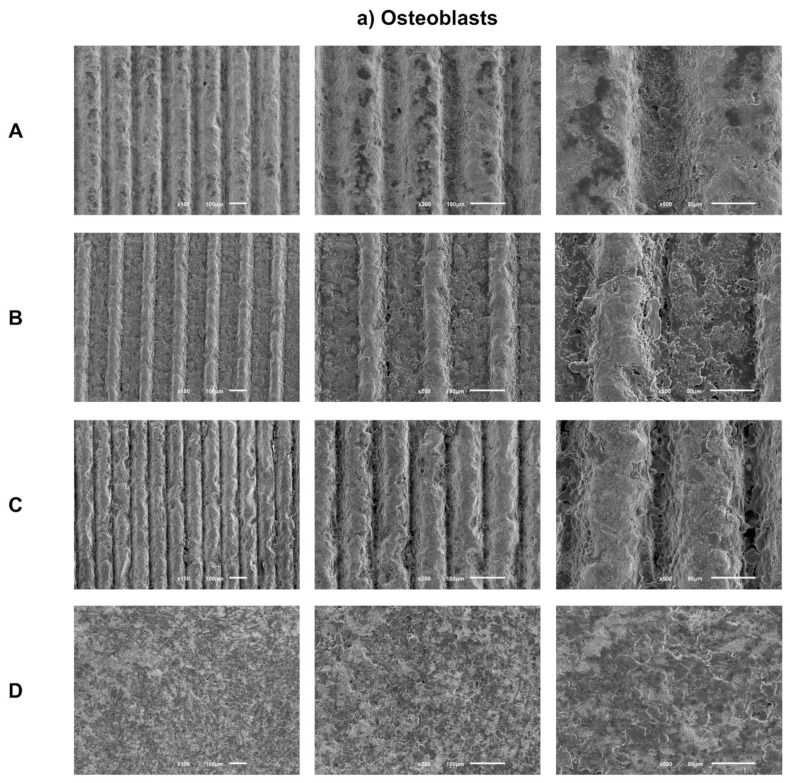
(**a**) Representative FEG-SEM micrographs of osteoblasts cultured on test samples and control surface after 1 day, with 100, 200, and 500× magnifications for each study group (as described in Table 1). (**b**) FEG-SEM micrographs of fibroblasts cultured on test samples and the control surface after 1 day, with 100, 200, and 500× magnifications for each study group (as described in Table 1).

**Figure 4 biomimetics-07-00049-f004:**
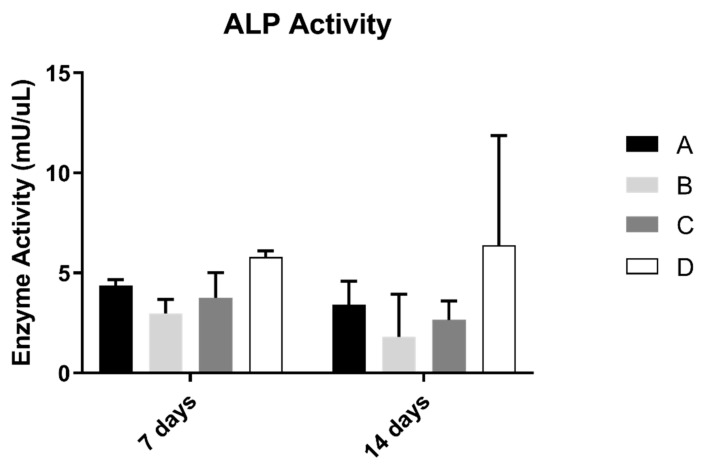
Mean alkaline phosphatase activity (mU/µL) measured in osteoblasts cultured for 7 and 14 days (N = 5), presented as the mean, for each study group (as described in Table 1). A Kruskal–Wallis test, followed by Dunn’s post-hoc tests, was used for comparison between study groups. Error bars represent the standard deviation. Statistical significance: *p*< 0.05.

**Figure 5 biomimetics-07-00049-f005:**
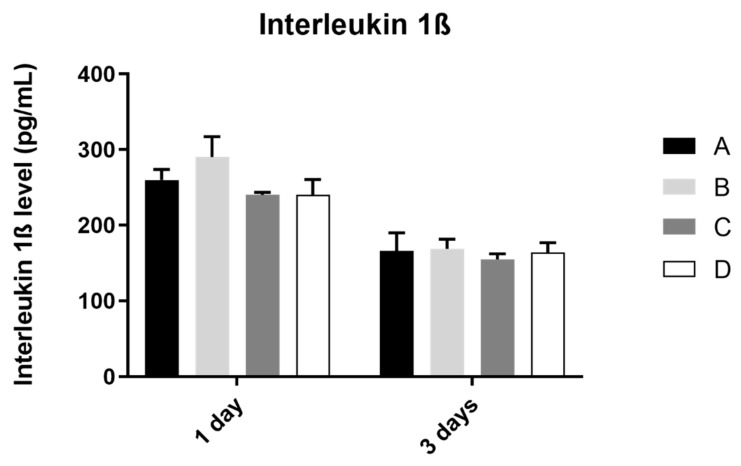
Mean concentration of interleukin 1ß (pg/mL) measured in osteoblasts culture in all groups at 1 and 3 days (N = 4), presented as mean and standard deviation, for each study group (as described in Table 1). The Kruskal–Wallis test, followed by Dunn’s post-hoc tests, was used for comparison between study groups. Error bars represent standard deviation. Statistical significance: *p*< 0.05.

**Figure 6 biomimetics-07-00049-f006:**
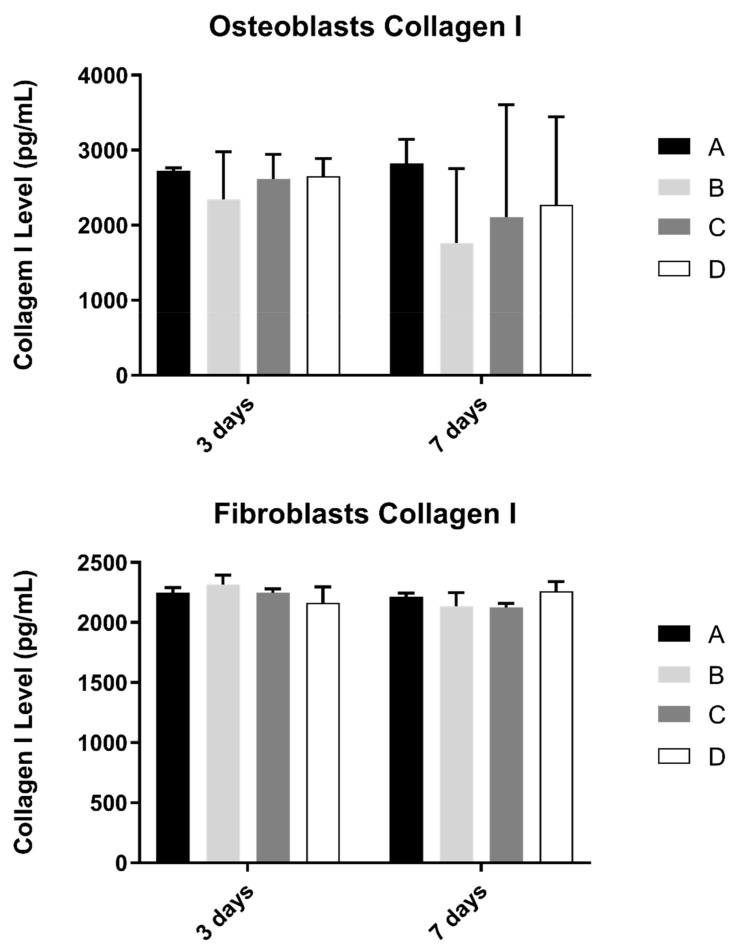
Collagen type I concentration (in pg/mL) obtained in osteoblasts and fibroblasts in all study groups at 3 and 7 days (N = 4), presented as mean and standard deviation (error bars), for each study group (as described in Table 1). The Kruskal–Wallis test, followed by Dunn’s post-hoc tests, was used for comparison between study groups. Statistical significance: *p* < 0.05.

**Figure 7 biomimetics-07-00049-f007:**
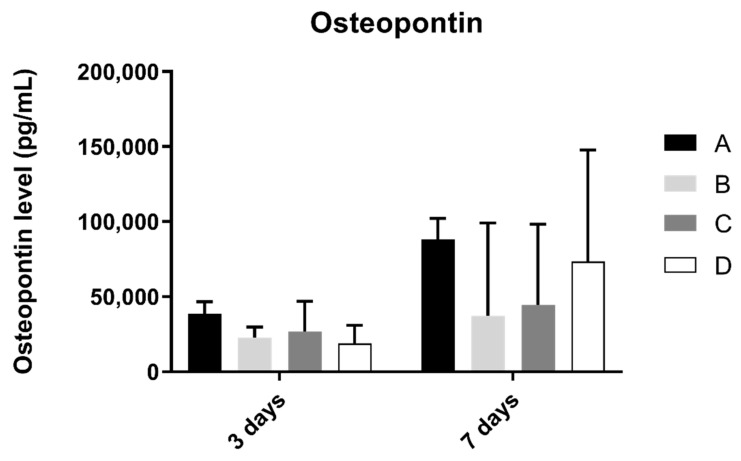
Osteopontin concentration (in pg/mL) obtained in osteoblast culture at 3 and 7 days of cell culture (N = 4), presented as mean and standard deviation (error bars), for each study group (as described in Table 1). A Kruskal–Wallis test, followed by Dunn’s post-hoc tests, was used for comparison between study groups. Statistical significance: *p* < 0.05.

**Figure 8 biomimetics-07-00049-f008:**
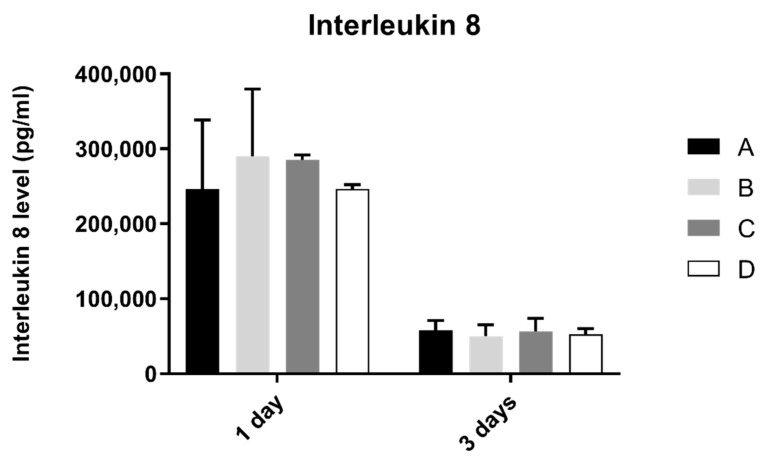
Interleukin 8 concentration (pg/mL) obtain in fibroblasts at 1 and 3 days (N = 4), presented as mean and standard deviation (error bars), for each study group (as described in Table 1). A Kruskal–Wallis test, followed by Dunn’s post-hoc tests, was used for comparison between study groups. Statistical significance: *p* < 0.05.

**Table 1 biomimetics-07-00049-t001:** A schematic table of the results obtained in terms of width and depth grooves by an innovative laser technique in μm (N = 10).

Group	Width (μm)	Depth (μm)
A	84.12 ± 5.13	36.35 ± 4.49
B	125.07 ± 5.29	23.01 ± 3.79
C	45.36 ± 2.37	50.54 ± 2.48
D (control)	No laser treatment	

## Data Availability

The authors declare that the data supporting the findings of this study are available within the article.

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
