# Peer review of "Modification of Zirconia Implant Surfaces by Nd:YAG Laser Grooves: Does It Change Cell Behavior?"

_biomimetics, 2022, doi:10.3390/biomimetics7020049_

Round 1

Reviewer 1 Report

The present study aimed to evaluate human osteoblasts and gingival fibroblasts response of textured Nd: YAG laser microgrooves, with different dimensions, on Zirconia implants surfaces. The subject is interesting and can contribute to the scientific literature. Some corrections are necessary to improve the text quality:

Title:

Separate the sentence with colons instead period.

Abstract:

Inform the disks’ material and dimension.

Use period to separate the decimal values from the numerical results.

Introduction:

Why an alternative method to the etching should be evaluated to treat zirconia implants? What are the disadvantages of acid etching in terms of time-consuming and dangerousness?

“It is noteworthy that there are few references associated with the Nd:YAG laser texturization of Zirconia implant surfaces and even less that evaluate the influence of different dimensions of textured grooves performed by this technique, so it still is not possible to extrapolate the real influence of these texturizations on cellular behavior.” Please, cite these few references.

An better explanation about the success-rate of zirconia implants should be provided as well as the association between surface treatment and osseointegration with zirconia.

Describe that the zirconia implant are promising alternative to replace titanium with adequate chewing load dissipation. Consider the following reference:

de Matos JD, Lopes GD, Nakano LJ, Ramos ND, Vasconcelos JE, Bottino MA, Tribst JP. Biomechanical evaluation of 3-unit fixed partial dentures on monotype and two-piece zirconia dental implants. Computer methods in biomechanics and biomedical engineering. 2021 Jun 22:1-8.

Describe your study’s hypothesis.

Methods:

How the Nd:YAG parameters were defined?

All samples were sandblasted? Including the Nd:YAG treated groups? If it is true, you are not evaluating the Nd:YAG effect but an association of surface treatments. Check if your control group is properly described.

Why the roughness was not measured? In your introduction, you state that “surface topography and roughness are important factors in the process of cellular contact”.

Discussion:

Describe if your study’s hypothesis have been rejected or not.

Improve your study’s limitation  description.

How the biofilm formation can be affected by the provided surface treatment?

What are the next steps in this research line?

Reviewer 2 Report

01

Where did the authors take the sandblasting-acid etching protocol from? How can one possibly know whether this protocol is the one considered as the “gold standard” and used by the main implant manufacturers?

02

The chemical composition of the surface is also extremely important to exert an influence on cell viability. Yet, this was not performed by the authors, especially considering that not every acid-etching protocol is able to sufficiently remove all alumina from the surface. This is a considerable flaw of this study. Hydrofluoric acid is a weak acid and Al2O3 is known to have some level of toxicity. That is why some implant manufacturers sandblast their surfaces with TiO2 particles instead.

03

The authors made use of 60 samples, but there were 2 groups and a series of different biological tests, which sometimes were further divided into sub-groups.

For instance, in the cell viability and proliferation assay of osteoblasts and fibroblasts, there were subgroups of osteoblasts and fibroblasts, further subdivided in days of observation. This would result in a small number of samples for each subgroup. Therefore, the authors can even ignore the normality test and use only non-parametric tests instead, as if the number of samples per group is low, the analysis calls for a non-parametric test regardless of the normality, as the normal distribution cannot properly be verified. The authors used ANOVA for this analysis, and I suspect (based on the supposed small number of samples for each group for this test) that they should have Kruskal-Wallis instead.

04

Some parts of the Discussion section consist of paragraphs beginning with a repetition of the results followed by the citation of the results of other studies, without an actual discussion of the findings of the study. As examples, the paragraphs beginning with:

“The ideal dimension for maximizing (…)”

“Regarding the effect of textured grooves width by (…)”

05

The limitations of the study were neither pointed out nor discussed.

Reviewer 3 Report

Abstract- The authors need to be clearer about what results refer to fibroblasts and what results refer to osteoblasts. These are different cells and it seems like the authors are comparing the cells, which are not, so it is a little misleading. The authors could also add a sentence on the important of implants in the beginning too.

Intro- Needs a few sentence on why gingival fibroblasts are important to implants. The authors are clear about why osteoblasts matter but I suggest the following references to help improve this:

Comparison of peri-implant and periodontal marginal soft tissues in health and disease (https://doi.org/10.1111/prd.12150)

And

Junctional epithelium and hemidesmosomes: Tape and rivets for solving the “percutaneous device dilemma” in dental and other permanent implants

(https://doi.org/10.1016/j.bioactmat.2022.03.019)

Methods- Either here or discussion the authors should add justification to the dimensions chosen.

Sample sizes for the number of ELISA wells, number of SEM micrographs taken, etc, etc need added somewhere.

Results- It is strange that proliferation did not go up over time. Can the authors comment why? Did the authors seed at too of an initial density to allow for growth to occur?

Scale bar labels are hard to read.

mL should be used, not ml in the figure labels on axises.

Discussion- There should be mentioned the important of IL1B and IL8.

um, not u – Line 303 on pg 12

Again, like the introduction, more information on soft tissue and fibroblasts and keratinocytes is needed. The references above can help, and others too.

Conclusion- OK

Other- Zirconia is sometimes capitalized and sometimes not; should not be.

Superscript numbers need to be superscript – this mistake is made a few times with XXX times 10^X. For example, pg 4, line 120.

Round 2

Reviewer 2 Report

“There were a total of 60 samples produced, divided in 4 groups (15 samples each – as described in table 1, enabling parametric analysis). The same samples were used in the various essays at multiple times of measurement.”

Fifteen samples in each group does not “enable” parametric analysis. It is still under the 30 samples per group. Moreover, even if normality can be reasonably assumed, in small samples tests that assume normally distributed data are likely to be underpowered to detect departures from the equal variance assumption. That is, use of these tests in small samples may lead researchers to incorrectly conclude that the equal variance assumption is justified. Therefore, Kruskal-Wallis with Dunn’s test would be more adequate.

For more information the authors can consult, for example, the following literature (which is not limited by these): (a) Altman DG, Gore SM, Gardner MJ, Pocock SJ. Statistical guidelines for contributors to medical journals. Br Med J (Clin Res Ed). 1983;286(6376): 1489–1493; (b) Fagerland MW. t-tests, non-parametric tests, and large studies--a paradox of statistical practice?. BMC Med Res Methodol. 2012;12: 78; (c) Morgan CJ. Use of proper statistical techniques for research studies with small samples. Am J Physiol Lung Cell Mol Physiol. 2017;313(5): L873–L877; (d) Dwivedi AK, Mallawaarachchi I, Alvarado LA. Analysis of small sample size studies using nonparametric bootstrap test with pooled resampling method. Stat Med. 2017;36(14): 2187–2205; (e) Wilcox R, Peterson TJ, McNitt-Gray JL. Data Analyses When Sample Sizes Are Small: Modern Advances for Dealing With Outliers, Skewed Distributions, and Heteroscedasticity. J Appl Biomech. 2018;34(4): 258–261.

Moreover, no power analysis was performed. Therefore, it is not possible to know the minimum number of samples in each group to distinguish the analyses of the results from true findings to pure chance.
